

# Protective effect of fustin against adjuvant-induced arthritis through the restoration of proinflammatory response and oxidative stress

Sultan Alshehri[1], Shareefa A. AlGhamdi[2,3], Amira M. Alghamdi[2], Syed Sarim Imam[1], Wael A. Mahdi[1], Mohammad A. Almaniea[1], Baraa Mohammed Hajjar[1], Fahad A. Al-Abbasi[2], Nadeem Sayyed[4] and Imran Kazmi[2]

[1] Department of Pharmaceutics, College of Pharmacy, King Saud University, Riyadh, Saudi Arabia
[2] Department of Biochemistry, Faculty of Sciences, King Abdulaziz University, Jeddah, Saudi Arabia
[3] Experimental Biochemistry Unit, King Fahd Medical Research Center, King Abdulaziz University, Jeddah, Saudi Arabia
[4] Glocal School of Pharmacy, Glocal University, Saharanpur, India

Corresponding author
Imran Kazmi, ikazmi@kau.edu.sa

## ABSTRACT

Rheumatoid arthritis causes irreparable damage to joints. The present research sought to check fustin's anti-arthritic efficacy against the complete Freund's adjuvant-induced arthritis paradigm in animals by altering the inflammatory response. In the rats, complete Freund's adjuvant was used to trigger arthritis and they received fustin at 50 and 100 mg/kg for 21 days. At regular intervals, the hind paw volume and arthritic score were assessed. After the trial period, hematological, antioxidant, pro-inflammatory cytokines, and other biochemical parameters were estimated. Fustin-treated rats showed the down-regulation of hind paw volume, arthritic score, and altered hematological parameters (TLC, DLC (neutrophil, lymphocyte, monocyte, eosinophil, basophil)). Furthermore, fustin significantly mitigates proinflammatory cytokine (reduced inter-leukin, tumor necrosis factor-a (TNF-$\alpha$), IL-6, IL-1$\beta$), oxidative stress (attenuated malondialdehyde (MDA), catalase (CAT), glutathione (GSH), superoxide dismutase (SOD)), attenuated production of prostaglandin E2 and myeloperoxidase (MPO) and improved nuclear factor erythroid 2-related factor (Nrf2) action. Fustin led to the benefit in arthritis-prone animals elicited by complete Freund's adjuvant *via* pro-inflammatory cytokine.

# INTRODUCTION

Rheumatoid arthritis (RA) is an autoimmune disease associated with the synovial joint lining and is a key cause of morbidity, mortality, and social hardship (*Fazal et al., 2018*). There are several pathological abnormalities associated with the immune structure that has a profound effect on the progression of diseases (*Nakayamada et al., 2016*). Inflammation of joints, synovial hyperplasia, swelling, and stiffness are the most obvious symptoms of rheumatoid arthritis (*Kumar et al., 2016*). Cells in the synovial intima are the primary cells

involved in the inflammatory reactions that ultimately lead to rheumatoid arthritis (*Guo et al., 2018*; *Okada et al., 2019*). Despite the fact that rheumatoid arthritis does not run in families, infection or environmental factors can trigger human RA genes under certain conditions, resulting in an autoimmune condition in which the immune structure triggers inflammation of the joints (*Yap et al., 2018*). In RA patients' whole blood and monocytes, there was a five-fold increase in the generation of mitochondrial ROS compared to normal patients, which implies that oxidative stress is a pathogenic feature of RA. Due to their crucial role as secondary messengers in the inflammatory and immunological cellular response in RA, free radicals are indirectly linked to joint destruction (*Ponist, Zloh & Bauerova, 2019*).

Experimenting with chronic disease rodent models allows for a better inclination of the physiopathological progression as well as the appraisal of potential new remedies. Adjuvant-induced arthritis (AIA) animal model has been established for decades to study the pathogenesis of arthritis, including rheumatoid arthritis (RA), gout, and osteoarthritis, and to evaluate the effectiveness of certain anti-arthritic drugs. Modeling of complete Freund's adjuvant (CFA)-elicited arthritis in animals has subsequently become popular in research to develop a treatment for rheumatoid arthritis and other chronic inflammatory arthropathies (*Jang et al., 2018*). The pathophysiology of rheumatoid arthritis is still unknown, but researchers have speculated that inflammatory intermediator, such as tumor necrosis factor-alpha, cyclooxygenase-2, and interleukin-6 are involved in the inflammatory response of synovial membranes and bone injury observed during rheumatoid arthritis (*Meehan et al., 2021*). In general, the researchers focused on inflammatory mediators, for the direction of rheumatoid arthritis in an animal model as well as in humans.

Plants contain polyphenolic compounds called flavonoids, which have been revealed to employ several attention-grabbing pharmacological sound effects. The active compound fustin appears to be derived from *Rhusverniciflua stokes*, which is a traditional herbal medicine from the Anacardiaceae family that is used for treating a variety of illnesses (*Li et al., 2020*). Among the diabetic model and high-fat diet model, fustin has been found to exhibit anti-antidiabetic, antihyperlipidemic, and antioxidant potential (*Gilani et al., 2021*). Fustin, on the other hand, ameliorates many cognitive deficits in diabetic animals (*Afzal et al., 2021*). Researchers have found that fustin has neuroprotective properties when exposed SK-N-SH dopaminergic neurons to 6-hydroxydopamine after pretreatment with fustin and this led to lower ROS levels and higher intracellular calcium levels ($Ca^{2+}$). Furthermore, a reduction in caspase-3 activity, Bax/Bcl-2 ratios, and p38 phosphorylation is also prevented by fustin (*Park et al., 2007*). Fustin's antiviral properties, exemplified by EC50 (91.2 to 197.3 µM) against viral hemorrhagic septicemia virus (VHSV) and hematopoietic necrosis virus (IHNV) (*Kang, Kang & Oh, 2012*). The most potent antibacterial activity was demonstrated by fustin against fish pathogens (*Jang et al., 2018*).

Patients prefer plant drugs to conventional medicines due to the unrelenting nature of the disease, terrible morbidity, ever-increasing medicinal costs, and trivial reaction to established drugs. Many plants that have traditionally been used by various tribal and rural cultures around the world have demonstrated a promising role as anti-arthritic agents. In recent years, phytocomponents have aroused considerable interest in the search for arthritis

treatment as they provide a plethora of active compounds including flavonoid compounds. Flavonoid-based therapies have been said to be safe and effective treating rheumatoid arthritis, and many are currently being studied (*Gautam et al., 2020*). Flavonoids show different activities, including antiviral, antiallergenic, anti-inflammatory, and vasodilation actions. However, of particular interest is the antioxidant activity of flavonoids because of their inherent ability to scavenge free radicals and reduce their formation (*Pier-Giorgio Pietta, 2000*). The objective of this study was to assess the anti-arthritic efficacy of fustin in animals by altering various parameters of Freund's adjuvant-induced arthritis. Fustin has some recently published pharmacological effects including protective effects on cognitive impairment in streptozotocin-induced diabetic rats (*Pietta, 2000*), antidiabetic properties in a high-fat diet, and streptozotocin-elicited diabetes in animals (*Afzal et al., 2021*), antihyperglycemics and antioxidants (*Gilani et al., 2021*) and anti-ulcerative activity on ethanol-induced gastric ulcers animal model (*Gilani et al., 2022*). For a more in-depth understanding, we evaluated interleukins IL-1 $\beta$, IL-6, tumour necrosis factor-a (TNF-a), catalase (CAT), glutathione (GSH), superoxide dismutase (SOD), malondialdehyde (MDA), prostaglandin E2, myeloperoxidase (MPO), alanine aminotransferase (ALT), aspartate aminotransferase (AST), alkaline phosphatase (ALP) and nuclear factor erythroid 2-related factor (Nrf2) in CFA-induced arthritis model.

## MATERIALS AND METHODS

### Animals

Male Wistar (10–12 weeks old; 180 $\pm$ 20 g) rats were procured from the animal center and research laboratory of T-G Services, Maharashtra, India. Under standard laboratory conditions, rats were placed in polypropylene cages at an ambient temperature of 25 $\pm$ 1 °C and of relative humidity 45–55%, with a 12:12 h light/dark cycle, and animals were included in the study with no previous procedure. Before the experiments, the rats were given seven days to adjust to the laboratory setting. These animals were given a free choice of commercial diet and tap water. The institutional animal ethics committee reviewed and approved the animal experimental research (IAEC/ TRS/PT/21/06) and research conducted as per the ARRIVE guideline (*Percie du Sert et al., 2020*).

### Chemicals

CFA was obtained from Sigma Aldrich (St. Louis, MO, USA). For the current experimental project, fustin (>98%, stability $\geq$4 years) received a gift sample from MSW Pharma, Maharashtra, India, and other chemicals were analytical grade and bought through an approved local vendor, chemical supplier, Maharashtra, India.

### Arthritis model

The present research sought to check fustin's anti-arthritic efficacy against the CFA-induced arthritis paradigm in laboratory animals. The animals ($N = 6$) were randomized and separated into four groups. A total of 24 experimental animals was utilized in this current research with the following;

    Group I: Normal control (received 5% DMSO)

Group II: Arthritic control (received 0.1 mL CFA intradermally)
Group III: Received fustin orally-50 mg/kg day soluble with DMSO for 21 days
Group IV: Received fustin orally-100 mg/kg day soluble with DMSO for 21 days

## CFA and fustin administration

All experimental groups received a 0.1 mL single dose of CFA injected intramuscularly into the right hind paw for provoking arthritis and received the oral administration of fustin (50 and 100 mg/kg per day) in the span of 21 days except normal control and arthritic control. On day 22nd, after the research protocol was accomplished, the rats were anesthetized with an intraperitoneal injection of 80 mg/kg ketamine and 10 mg/kg xylazine. Orbital blood samples were also obtained for biochemical analysis. The rats were sacrificed by cervical dislocation (*Aloke et al., 2021*).

## Appraisal of arthritis

### Hind paw volume

At regular intervals (on 7, 14, and 21 days), all experimental animals' hind paw volumes were measured as visible indicators of arthritis modalities. For all the experimental groups, paw volume changes were measured with a plethysmometer (Orchid, Nungambakkam, India). A plethysmometer was used to estimate the edematous hind paw of an inflamed rat.

### Arthritic score

The observer scored visual arthritic changes every 7 days from day 0 to the termination of the study. The intensity of arthritis in the paws of rats was assessed and ranked from 0 to 4. Grade 0 denotes no puffiness; grade 1 shows erythema or slight swelling on the paw finger; grade 2 denotes one or more fingers paw swelling; grade 3 denotes wrist or ankle swelling; and grade 4 denotes drastic arthritic swelling in the wrist and fingers. The highest arthritic valuation fixed for rats induced with CFA is an 8 score. An observer who was not aware of the experimental protocol took visual arthritic measurements (*Hussain et al., 2021*).

### Hematological factors

On day 22, blood samples were collected *via* the retroorbital plexus with hematocrit capillary tubes. A blood cell counter was used to measure hematological parameters such as total leukocyte count (TLC) and differential leukocyte count (neutrophil, lymphocyte, eosinophil, monocyte, basophil).

### Biochemical parameters

Biochemical considerations *i.e.,* ALP, AST, and ALT were estimated by commercial kits and manufacture directions (Modern Lab, Maharashtra, India) (*Kamal et al., 2021*).

## Prostaglandin E2 (PGE2)

Blood serum was collected from the rats on day 22 and stored at 80 °C before being assayed. PGE2 levels in the serum were determined using ELISA kits by the manufacturer's recommendations (MSW Pharm, Maharashtra, India). The amounts of PGE2 were estimated as ng/mL (*Chen et al., 2014*).

### Proinflammatory cytokines

Standard ELISA assay kits (MSW Pharm, Maharashtra, India) were utilized to estimate the protein appearance of proinflammatory cytokines in serum such as TNF-$\alpha$, IL-1 $\beta$, and IL-6 (*Saleem et al., 2020*).

### Endogenous (enzymatic and non-enzymatic) antioxidants and oxidative stress parameters

Standard methods were used to measure the serum activities of SOD, GSH, CAT, and MDA to identify the locations of oxidative stress indicators in arthritic animals (*Misra & Fridovich, 1972*; *Sinha, 1972*; *Moron, Depierre & Mannervik, 1979*).

### Nrf2 and MPO

The paw samples were homogenized in potassium phosphate buffer (20 mmol/L, pH 7.4) with 0.1 mmol/L ethylenediaminetetraacetic acid and centrifuged at 2,000 g for 10 min at 4 ° C. $H_2O_2$ in phosphate buffer (50 mM, pH 6) was used as the substrate in the method defined by *Bradley et al. (1982)* to find out MPO activity in tissue homogenates. Assay buffer (7.5 mg o-dianisidine-HCl, 5 mL 0.0005% $H_2O_2$ in 40 mL phosphate buffer) was prepared. To 280 µL assay buffer, 20 µL tissue homogenate was added. MPO activity was measured kinetically for 5 min at 460 nm. The *Lal et al. (2021)* method was used to calculate the nuclear factor erythroid 2-related factor (Nrf2). Serum Nrf2 levels were quantified in the biochemistry laboratory using the enzyme-linked immunosorbent assay (Sigma-Aldrich, St. Louis, MO, USA) and read at 450 nm on a microplate reader.

## Statistical analysis

The findings are reported as mean ± SEM with one-way ANOVA followed by Tukey's *post hoc* test for statistical expressive style (GraphPad Prism version 8.0.). Scoring data was analyzed statistically *via* the Kruskal–Wallis non-parametric test.

## RESULTS

### Changes in hind paw volume

Fustin's effect on hind paw volume in CFA trigger arthritis animals is displayed in Figs. 1A–1C. Throughout the experiment, arthritis control rats were significantly ($P < 0.001$) a linear rise in paw volume as compared to normal control rats. On 7, 14, and 21 days, fustin 50 and 100 mg/kg inhibited inflammatory hind paw volume significantly reduced than the arthritis control group. The paw volume of rats decreased as the dose of fustin was increased, indicating that paw volume dropped significantly based on dose (F $(3, 60)$ = 91.50, ($P < 0.0001$)). Fustin 100 mg/kg recorded the maximum percentage of inhibition.

### Arthritic score

The scoring system depicted in Figs. 2A–2C was used to appraise the pattern of arthritis improvement. The arthritic score was significantly amplified during arthritis, confirming the disease advancement. In the CFA-induced arthritic rats, a similar effect was ascertained. On day 21, arthritis control rats had a significantly greater arthritic index score than normal control animals. No significant difference was detected between the treatment groups
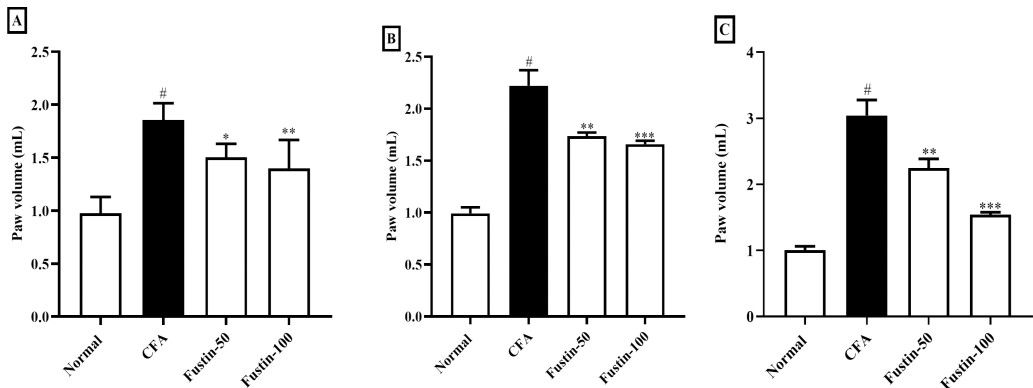

**Figure 1 Effect of fustin on hind paw volume in CFA-induced arthritis model ($n = 6$).** (A) 7th day (B) 14th day (C) 21th day. # $P < 0.001$ *vs.* Normal, \*\*\*$P < 0.0001$, \*\*$P < 0.001$, \*$P < 0.05$ *vs.* CFA.

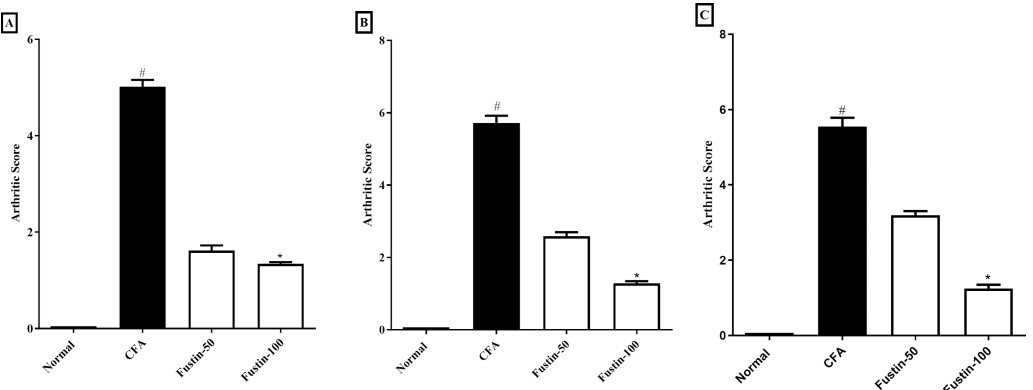

**Figure 2 Effect of fustin on arthritis score in CFA-induced arthritis model ($n = 6$).** (A) 7th day (B) 14th day (C) 21th day. # $P < 0.001$ *vs.* Normal, \*$P < 0.05$ *vs.* CFA.

compared to the normal control group. Using Kruskal–Walli's test, fustin 50 and100 mg/kg to the disease control at different time intervals exhibited an evidentiary depletion in arthritic index score at the different time intervals (F (3, 60) = 985.9, ($P < 0.0001$)) (Figs. 2A–2C) as a comparison to arthritis control.

## Hematology profile

As illustrated by a rise in total lymphocyte count and differential count (neutrophil, lymphocyte, monocyte, eosinophil, basophil), the arthritic control rats had substantial ($P < 0.001$) leukocytosis compared with the normal control rats. No discernible difference between the therapy groups and the normal control group was found. However, in demarcation to the arthritic rats, the CFA-induced arthritic rats boost with fustin at both doses indicated an evidentiary drop-off in TLC (F (3, 20) = 62.19, ($P < 0.0001$)), as well as a significant decrease in lymphocyte (F (3, 20) = 13.94, ($P < 0.0001$)), monocyte (F (3, 20) = 25.81, ($P < 0.0001$)), neutrophil (F (3, 20) = 29.10, ($P < 0.0001$)), eosinophil (F

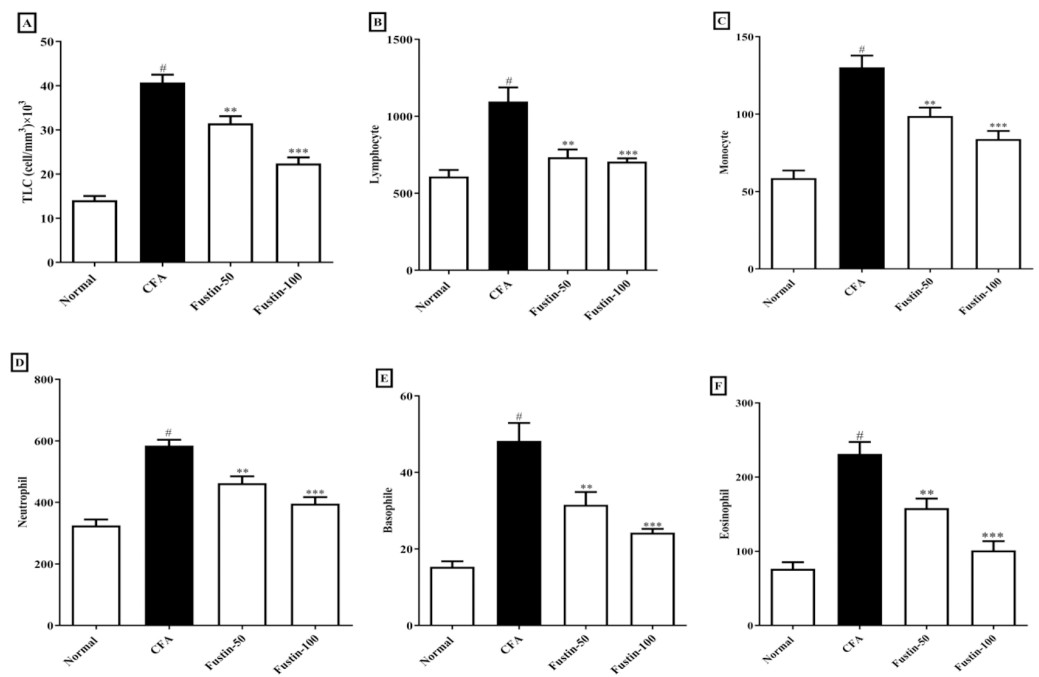

**Figure 3** Effect of fustin on (A) total leukocyte count (B) lymphocyte count (C) monocyte count (D) neutrophil count (E) basophile count (F) eosinophil count in CFA-induced arthritis model ($n = 6$). # $P < 0.001$ *vs.* Normal, ***$P < 0.0001$, **$P < 0.001$ *vs.* CFA.

(3, 20) = 29.01, ($P < 0.0001$)), and basophil counts (F (3, 20) = 21.62, ($P < 0.0001$)) as compared to arthritis control (Figs. 3A–3F).

## Biochemical profile

When rats treated with an arthritis model had higher ($P < 0.001$) levels of the hepatic damage markers ALT, AST, and ALP than healthy controls, they were classified as having liver damage. There was no significant difference between the treatment groups and the normal control group. Figures 4A–4C shows that rats were given fustin at 50 and 100 mg/kg significantly reduced serum ALT (F (3, 20) = 16.43, ($P < 0.0001$)), AST (F (3, 20) = 16.62, ($P < 0.0001$)), and ALP (F (3, 20) = 36.39, ($P < 0.0001$)) than arthritis control rats.

## Prostaglandin E2 (PGE2)

Figure 4D illustrates that the serum PGE2 concentrations of arthritis rats dramatically increased ($P < 0.001$) when compared to the normal control rats. No noticeable difference between the fustin treatment 50 and 100 mg/kg groups and the normal control rats was found. When contrasted to the arthritis control rats, fustin 50 and 100 mg/kg significantly reduced PGE2 concentrations (F (3, 20) = 20.83, ($P < 0.0001$)).

## Proinflammatory cytokines

Figures 4E–4G shows the proinflammatory cytokines such as TNF-$\alpha$, IL-1 $\beta$, and IL-6 in contrastive study groups. No statistically significant difference between the normal rats and 50 and 100 mg/kg treatment groups rats. When serum levels of TNF-$\alpha$, IL-1 $\beta$, and

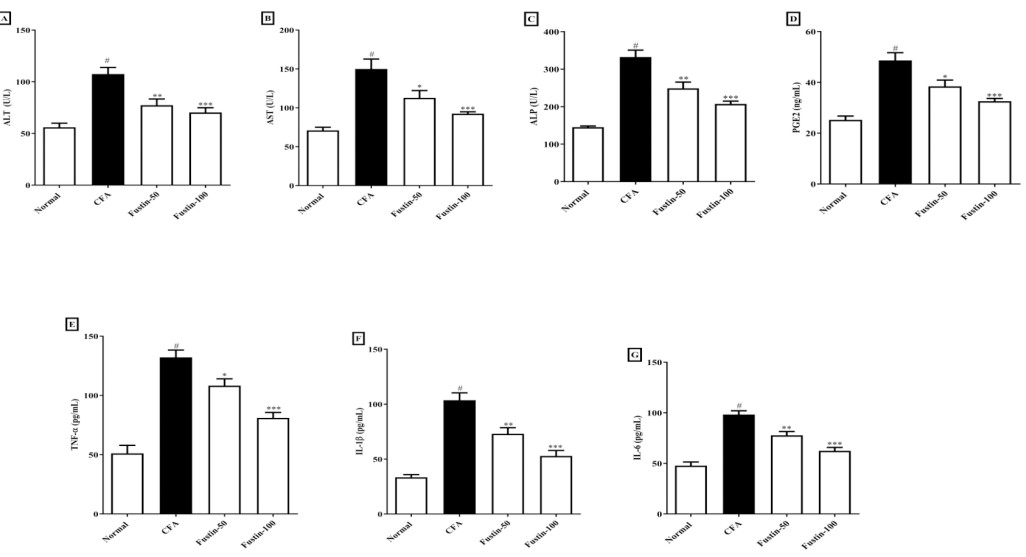

**Figure 4** Effect of fustin on (A) ALT (B) AST (C) ALP (D) prostaglandin E2 (E) TNF-$\alpha$ (F) IL-1$\beta$ (G) IL-6 in CFA induced arthritis model ($n = 6$). # $P < 0.001$ *vs.* Normal, ***$P < 0.0001$, **$P < 0.001$, *$P < 0.05$ *vs.* CFA.

IL-6 were examined in arthritis and normal control rats, there was a significant rise in given proinflammatory cytokines in arthritic disease rats ($P < 0.001$). One-way ANOVA followed by Tukey's *post hoc* test, upon supplementation with fustin at 50 and 100 mg/kg, the serum levels of TNF-$\alpha$ (F (3, 20) = 35.44, ($P < 0.0001$)), IL-1 $\beta$ (F (3, 20) = 33.82, ($P < 0.0001$)), IL-6 (F (3, 20) = 36.04, ($P < 0.0001$)) were pronounced brought down in fustin treatment rats, when related to the untreated counterparts.

## Antioxidant enzymes and oxidative stress parameters

The status of oxidative stress can be assessed by looking at antioxidants in CFA-induced arthritic animals. When compared to normal healthy control rats, CFA-induced animals had significantly lower antioxidant enzymes CAT, SOD, and GSH. No significant comparison was detected in the CAT, SOD, and GSH in the fustin treatment groups compared to the normal control group. One-way ANOVA followed by Tukey's *post hoc* test, fustin treatment groups demonstrated significant improvement in CAT (F (3, 20) = 21.54, ($P < 0.0001$)), SOD (F (3, 20) = 33.31, ($P < 0.0001$)) and GSH (F (3, 20) = 10.60, ($P < 0.0001$)) compared with rats fed on CFA alone (Figs. 5A–5C). Figure 5D demonstrates the effects of fustin on MDA levels in the serum of various experimental animals. MDA levels were found to increase in arthritis animals when MDA levels were measured in different experimental groups. There was no meaningful difference between the normal rats and the treatment group rats. Fustin 50 and 100 mg/kg treatment expressively, decreased MDA range in treated rats when related to CFA alone administered animals (F (3, 20) = 28.65, ($P < 0.0001$)).

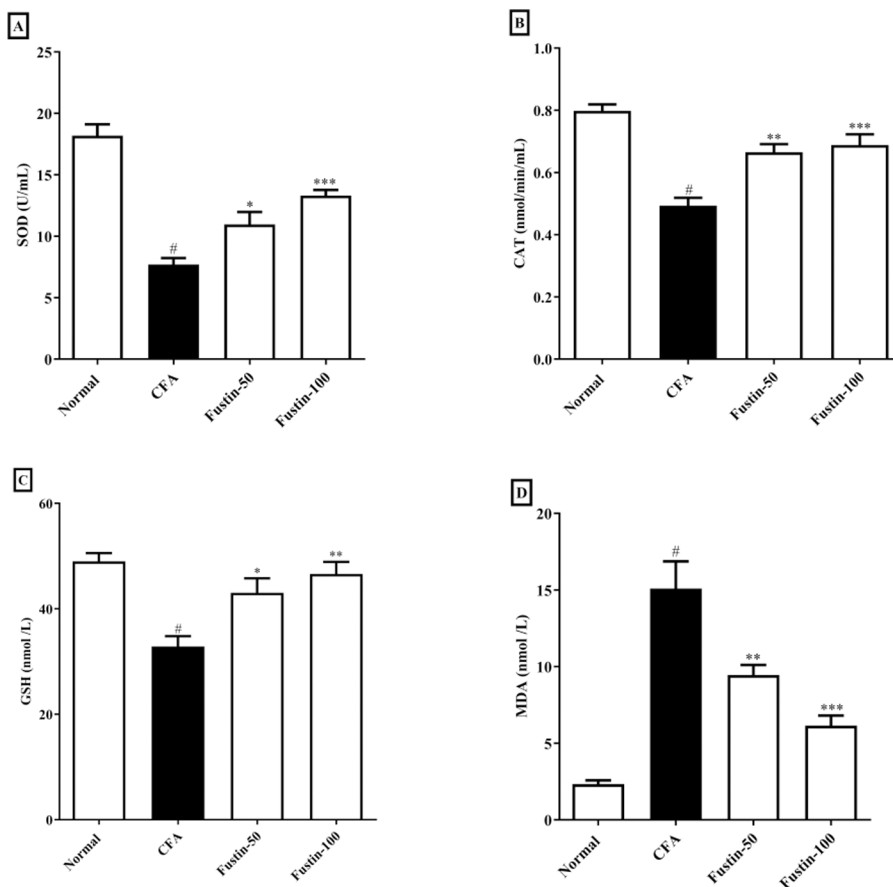

**Figure 5** **Effect of fustin on (A) SOD (B) CAT (C) GSH (D) MDA in CFA induced arthritis model ($n =$ 6).** # $P < 0.001$ *vs.* Normal, *** $P < 0.0001$, ** $P < 0.001$, * $P < 0.05$ *vs.* CFA.

## MPO and Nrf-2

After injection of CFA, MPO a marker of neutrophil infiltration, which also plays a role in oxidative impairment in rheumatoid arthritis, was significantly elevated in joint tissues. In arthritis control rats, fustin administration significantly reduced these elevations (F (3, 20) = 30.19, ($P < 0.0001$)). When compared to the normal control group, no detectable difference between the treatment centers was seen. Nrf-2 modulate the appearance of a variety of antioxidant consequence element-dependent genes at both the basal and induced levels. Remarkably, the Nrf-2 expression was reduced level in the disease control experimental rats, whereas the expression of Nrf-2 was dramatically improved in the fustin-treated rats, implying that the antioxidant defense system had been activated (F (3, 20) = 24.20, ($P < 0.0001$)) (Figs. 6A–6B). There was absolutely no meaningful difference between the normal and treatment groups of animals.

## DISCUSSION

Rheumatoid arthritis is the supreme leading cause of frailty, touching billions of people worldwide (*Patel et al., 2021*). The possible adverse effects of currently available treatments

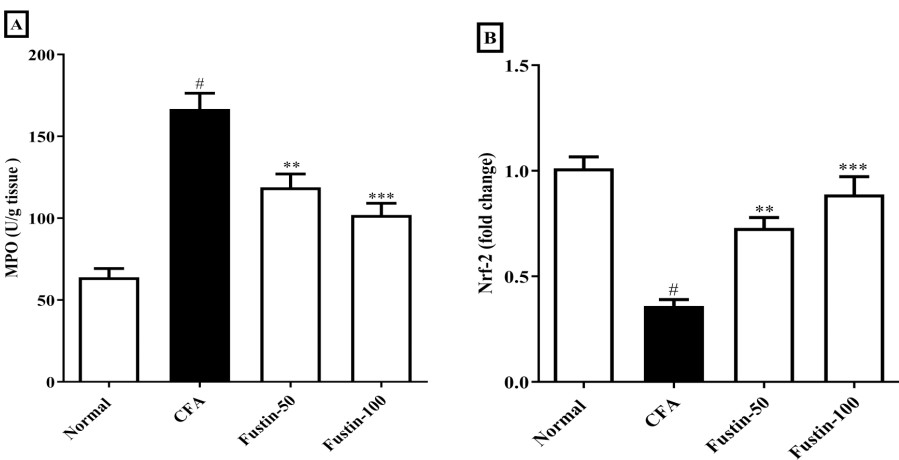

**Figure 6** **Effect of fustin on (A) MPO (B) Nrf-2 in CFA-induced arthritis model ($n = 6$).** # $P < 0.001$ vs. Normal, *** $P < 0.0001$, ** $P < 0.001$ vs. CFA.

necessarily entail the development of more operational medical aid that will acquire a broader scope of patients. For decades, the adjuvant-induced arthritis (AIA) animal model has been used to study the pathogenesis of arthritis, including rheumatoid arthritis, gout, and osteoarthritis, as well as the efficacy of various anti-arthritic drugs. *Stoerk et al. (1954)* pioneered the CFA-induced arthritic model, which has since been extensively modified for studies on acute or chronic, mono- or polyarthritis, and inflammatory mechanisms. Various CFA doses were tested in the past to establish arthritic rat models *via* subcutaneous, intradermal, or intraarticular routes. Few studies, however, included nociceptive and inflammatory effects in determining the optimal CFA dose for developing a chronic polyarthritic rat model that mimicked RA in humans. Fustin anti-arthritic potential in CFA-induced arthritis experimental rat model was investigated by altering the inflammatory response and the antiarthritis activity was established through measurement of improved paw thickness, arthritis score, and altered various hematological, biochemical, proinflammatory, oxidative stress, prostaglandin E2 level and neutrophil infiltration markers, which characterize an arthritic process. According to previous literature evidence, animals induced with rheumatoid arthritis showed an immunological and pathological comeback that was parallel to that of rheumatoid arthritis in human beings (*Ferraccioli & Zizzo, 2011*; *Bersellini Farinotti, 2019*).

Using a plethysmometer to measure paw thickness during CFA-induced arthritis is a well-established and standardized method (*Pradhan & Singh, 2021*; *Kumar & Rai, 2012*). A plethysmometer was used to estimate the edematous hind paw of an inflamed rat in the current study. Marked swelling, joint discomfort, and edema detected in the hind paws of rats after intradermal CFA might be the consequence of persisting inflammatory activity (*Fan et al., 2012*). Joint swelling can indicate the procession of rheumatoid arthritis in rats (*Hawkins et al., 2015*). Synovial edema develops in experimental arthritic rats induced by CFA as a result of an increase in synovial fluid production and a vascular infiltration into the inflammatory region (*Uttra & Hasan, 2017*). Our studies revealed increased edema

and penetration into cells in rats induced with rheumatoid arthritis, both of which signal chronic inflammation. When fustin was given to trial rats, it resulted in a step-down in arthritis score and paw volume in experimental rats challenged with CFA. Fustin treatment reduced paw thickness, possibly due to inflammatory mediator inhibition, implying that it has anti-inflammatory properties against CFA-induced arthritis.

Multiple hematological changes are also linked to rheumatoid arthritis. In the arthritic control animals, TLC and DLC data revealed severe leukocytosis. Furthermore, the increase in total leukocyte count in CFA-faced arthritic animals indicates the stimulation of immune structures against the invasive infective agent, which releases IL-1 $\beta$ and TNF-$\alpha$ (*Kim et al., 2016*). These findings are supported by histological evidence of joint capsule inflamed diffusely, lymphoid hyperplasia in the spleen, and mononuclear leukocytosis in the thymus of CFA-induced arthritic animals (*Ahmed et al., 2018*). Fustin treatments resulted in a significant reduction in elevated total leukocyte as well as differential lymphocyte counts, indicating that the treatments have a survival benefit against CFA-induced arthritis.

CFA models may have altered liver enzymes indicative of liver function due to the increased hepatic and bone fractions. This is linked to periarticular osteopenia and localized bone erosion (*Singh et al., 2021*). Fustin treatment of arthritic rats significantly attenuate raised ALP, AST, and ALT in experimental rats. The fact that the raised AST, ALT, and ALP levels remarkably declined proposed that the fustin treatment did not cause hepatic damage in the experimental rats. Previous research has shown that fustin has hepatoprotective effects in HFD (high-fat diet) streptozotocin-prompted diabetic rats and HFD-induced nonalcoholic fatty liver (*Lee et al., 2020*).

According to numerous studies, synovial cells, monocytes, and macrophages that infiltrate the synovial tissue secrete a variety of pro-inflammatory cytokines. Three of the most essential aspects in the onset and regression of rheumatoid arthritis are TNF-$\alpha$, IL-1 $\beta$, and IL-6 (*Yap et al., 2018*). As an outcome, pro-inflammatory cytokines have emerged as promising targets for rheumatoid arthritis. TNF-$\alpha$, IL-6, and IL-1 $\beta$ production were all tempered expressively in CFA-induced arthritis rats given fustin in our study. Fustin's anti-arthritic effect is linked to a reduction in proinflammatory cytokines, according to these findings. Phagocytes produce too much TNF-$\alpha$, which stimulates leucocyte adhesion and penetration with the vascular endothelium, causing inflammation. Furthermore, TNF-$\alpha$ suppresses bone collagen synthesis primarily by increasing bone destruction, and fibroblast hyperplasia (*Karwasra et al., 2021*). Likewise, oxidative stress is a major promoter of inflammation. Our findings suggest that fustin has anti-inflammatory properties that could be used to treat inflammation-related arthritis.

TNF-$\alpha$ accumulation has been shown to exaggerate the countenance of PGE2, IL-1 $\beta$, and IL-6 resulting in synovial joint hyperplasia, increased accretion of enzymes causing devastation, triggering of collagenase, and osteoclast affliction, all of which contribute to arthritic decay (*Alunno et al., 2017*). In CFA-induced rats, increased PGE-2 levels were found to be closely related to joint swelling, redness, pain, blood vessel dilatation, and cartilage corrosion. Similarly, fustin with antiarthritic effects must inhibit the expression of pro-inflammatory modulators in a direction to reduce the arthritis severity. Thus, the current study conclusions also revealed that fustin, when administered to CFA-induced

arthritic rats, resulted in an evidentiary decline in prostaglandin E2 anti-inflammatory modulators. We measured myeloperoxidase activity in rat neutrophils regularly as a marker for the inflammatory state. Many natural compounds work in the same way that NSAIDs do by inhibiting inflammatory pathways. As a result, the fustin anti-inflammatory effect could be attributed to decreased inflammatory cell infiltration. The current study found that reduces anti-oxidant markers (SOD, CAT, and GSH) while increasing pro-oxidant markers (MDA) in arthritic control rats compared to healthy rats. Fustin restores the disparity between pro-oxidant and antioxidant markers, implying that it may play a role in retreating the destruction caused by free radicals. In arthritis-induced rats complemented with fustin, the prominence of antioxidant enzymes such as GSH, CAT, and SOD was overexpressed, and the eminence of lipid peroxidation was decreased, confirming the antioxidant activity. Nrf2 is a critical transcriptional controller that mediates oxidative stress protection. When exposed to oxidative stress, Nrf2 leaves Keap1 and translocates into the nucleus, where it impasses to antioxidant response elements (AREs) in promoter regions, regulating antioxidant expression, and creating inducible antioxidant defense (*Gureev, Shaforostova & Popov, 2019*). In the current study, CFA-induced nuclear translocation of Nrf2 and reduced levels were observed, indicating that the Nrf2 signaling was activated by CFA. However, the expression of Nrf-2 was dramatically increased in fustin-treated rats. This result agreed with the previous report (*Choi et al., 2014*). Fustin inhibited proinflammatory cytokine pathway (IL-1 $\beta$, TNF-$\alpha$), oxidative stress, production of prostaglandin E2 and myeloperoxidase (MPO) as well as improved endogenous antioxidants (CAT, GSH, SOD) and Nrf2 action in CFA induced rats by modulating various signaling pathways and transcription factors involved in inflammation and oxidative stress. The underlying molecular mechanisms of action are most likely involved in fustin's capacity to exert potent antioxidant effects and its flavonoid factors by controlling the aforementioned pathway in the CFA-induced arthritis paradigm. Further studies are needed to confirm these mechanisms and determine the optimal dose and duration for fustin treatment to achieve its anti-inflammatory and antioxidant effects. A limitation of our study is that we have not to study western blot, immune staining of the slices for the arthritis biomarkers, and comparable efficacy with standard antiarthritic drugs. The acute arthritis rat model of arthritis is insufficient for studying the long-term safety and toxicity profile. As a result, evaluating a chronic arthritic model would be useful in determining the effect of longer exposure and the presence of reversible side effects. More exploration will be required in the future to perform mechanistic study and its application in therapeutic circumstances.

## CONCLUSION

In this study, we attempted to assess fustin effect on the CFA persuasion arthritis paradigm in experimental animals by inhibiting inflammatory mediators. The beneficial effects of fustin were mediated through lowering inflammation mediators and oxidative stress. Enhancements in hematological and biochemical markers may guard against the disease

conditions. Further research, including the presence of rheumatoid factor and C-reactive protein should be carried out to confirm the model's similarities to RA in humans.

### Funding
This work was supported by the Deanship of Scientific Research, King Saud University, Riyadh, Saudi Arabia. The funders had no role in study design, data collection and analysis, decision to publish, or preparation of the manuscript.

### Grant Disclosures
The following grant information was disclosed by the authors:
Deanship of Scientific Research, King Saud University, Riyadh, Saudi Arabia.

### Competing Interests
The authors declare there are no competing interests.

### Author Contributions
- Sultan Alshehri analyzed the data, authored or reviewed drafts of the article, and approved the final draft.
- Shareefa A. AlGhamdi analyzed the data, authored or reviewed drafts of the article, and approved the final draft.
- Amira M. Alghamdi analyzed the data, authored or reviewed drafts of the article, and approved the final draft.
- Syed Sarim Imam analyzed the data, authored or reviewed drafts of the article, and approved the final draft.
- Wael A. Mahdi analyzed the data, authored or reviewed drafts of the article, and approved the final draft.
- Mohammad A. Almaniea analyzed the data, authored or reviewed drafts of the article, and approved the final draft.
- Baraa Mohammed Hajjar analyzed the data, authored or reviewed drafts of the article, and approved the final draft.
- Fahad A. Al-Abbasi analyzed the data, authored or reviewed drafts of the article, and approved the final draft.
- Nadeem Sayyed performed the experiments, prepared figures and/or tables, and approved the final draft.
- Imran Kazmi conceived and designed the experiments, authored or reviewed drafts of the article, and approved the final draft.

### Animal Ethics
The following information was supplied relating to ethical approvals (i.e., approving body and any reference numbers):
  Institutional Ethical Committee (IAEC/ TRS/PT/21/06)

## Data Availability

The raw data are available in the Supplementary File.

## Supplemental Information

Supplemental information for this article can be found online at http://dx.doi.org/10.7717/peerj.15532#supplemental-information.

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
