# Peer review of "Protective effect of fustin against adjuvant-induced arthritis through the restoration of proinflammatory response and oxidative stress"

_PeerJ, doi:10.7717/peerj.15532_

## Round 0.1 · original submission · Major Revisions

Please amend your manuscript according to the reviewers' comments.

One of the reviewers noted that CFA (complete Freund's adjuvant) is generally considered a harsh treatment to animals. Please provide a copy of your ethical approval document with the next revision.

Reviewer 1 ·

Basic reporting

This study aimed to assess the effect of fustin on the complete Freund's adjuvant persuasion arthritis murine model by inhibiting inflammatory mediators.

The language used in the article is generally clear, unambiguous and professional. However, there are still some places can be improved.
Line 171 "50,100 mg/kg" can be misleading. The authors possible wanted to say "50 and 100 mg/kg"
Line 372 "complete freund's adjuvant" better to be "complete Freund's adjuvant"

The figures enclosed at the end of the manuscript are not the same as the ones in the article. The authors should check that. Also the numbering of the figures are different.

The author should also consider combining some of the figures. In addition, the fonts of Fig. 2A-C are different from other figures. It would be better to use the same fonts.

Figure 5B (enclosed at the end) is missing, there are two Figure 5A.

The result part can be elaborated more to give an idea to the readers about the meaning of the findings.

Experimental design

The research question is well-defined, relevant and meaningful.

The methods can be described better. A lot of the details are still missing.
For example, line 134 the information for the blood collection tube is missing.
Line 141, the information of microtiter plate is missing. The information for the kit is missing.
Also the information for the optical density calculation is missing (Line 145).
The detail of the methods part needs to be checked.

Line 138, AST is mentioned twice in "AST and AST".

Validity of the findings

The finding is meaningful to further developing the use of fustin. It would be better if the authors can discuss more in the manuscript about the potential clinical use in the future.

Reviewer 2 ·

Basic reporting

This study focuses on use of plant metabolite fustin as a potential therapeutic solution to treat rheumatoid arthritis, using the CFA-induced rheumatoid arthritis model in rats. The authors demonstrated that fustin daily administration across a span of up to 21 days showed significant efficacies to revert symptoms and almost all common molecular biomarkers of RA. The article is well written despite a few typos and organized with clear logical flow. All figures were with high quality and described well for understanding. Raw data for key figures are included and clearly labeled as well.

Experimental design

This study shows the novel therapeutic solution, plant-based biomolecule fustin in treatment for RA. The authors first used CFA compounds as used in previous studies to induce RA in rats. They then demonstrated that fustin daily administration across a span of up to 21 days showed significant efficacies to revert symptoms by relevant measurement methods. After drug administration cycles, the authors also took thorough blood test from the 6 experimental rats and observed almost all common molecular biomarkers of RA reverted by fustin, validated by biochemical assays like ELISA.

Validity of the findings

Overall the experiments were designed well and accompanied with reasonable controls. The statistical analyses were relevant and solid to support the authors hypotheses.

Additional comments

Major comments:
1. The authors described the use of male rats as experimental subjects. Why was this restricted to males and would the authors expect differential results if for female subjects?
2. The study showed that for some immunomarkers, fustin almost reverted the RA samples 100% back compared to the healthy state. Yet for other markers and for some symptoms like paw volume, fustin only partially converted the disease states back to normal. The authors may need to provide hypothesis in the discussion on why these discrepancies for fustin efficacy exists when looking at different markers and how this may guide the future optimization of treatment strategies. For example, would any prolonged treatments from fustin may be needed for a more thorough cure?
3. The authors need to address if any of the effects of fustin were against the artificial off-target effects from CFA itself, and not the RA disease pathways. To this end, similar tests with fustin treatment on RA models induced in ways other than CFA may be needed, or at least discussed upon.
Minor comments:
1. The manuscript is with occasional typos, for example, in line 144, the volume supposedly should be 50 µL instead of 50 mL.
2. The authors are recommended to add in short rationales for experimental designs before diving into data details in the results section.
3. For clarification, the authors are recommended to provide a quick summary or specify any modifications of existing protocols in the material and methods section, rather than simply citing the references.
4. Some figures can be merged as panels in the same figure if they’re referenced as a whole in the main text, such as figures 5-7.

Reviewer 3 ·

Basic reporting

In their work, authors evaluated anti-arthritis effect of fustin against complete Freund’s adjuvant-induced arthritis paradigm in animals. The hind paw volume and arthritic score were assessed. Then, Proinflammatory and stress oxidative biomarkers were assayed. Fustin has shown an interesting anti-arthritis activity. However, the manuscript has to be revised.

Experimental design

This research work suits the scope of the journal. The research question is well defined and meaningful. All the methods are reproducible.

Validity of the findings

The idea of evaluating the anti-arthritis activity of fustin is not new, as it has already been tested by Choi et al. (2003) at a dose of 30 mg/kg. The results showed a significant effect of fustin, which is in agreement with your results. The interest of your findings lies in the fact that you have tested other doses and measured other biomarkers that could be involved in the pathogenesis of arthritis, thereby strengthening your argument. However, the statistical data should be revisited.


Choi J, Yoon B-J, Han YN, Lee K-T, Ha J, Jung H-J, et al. Antirheumatoid Arthritis Effect of Rhus verniciflua and of the Active Component, Sulfuretin. Planta Med 2003;69:899–904. https://doi.org/10.1055/s-2003-45097.

Additional comments

Introduction
1- use direct, concise sentences: for example, line 46: Rheumatoid arthritis is an autoimmune disease associated with…instead of ‘An autoimmune disease, rheumatoid arthritis (RA) is principally associated with …’
2- The introduction needs to be well restructured for a logical flow of ideas: You presented the importance of plants flavonoids and pharmacological properties of fustin from Line 56 to 67, then from line 83 to 93. Between these two paragraphs you talked about your animal model and the importance of natural substances in medicine. This transition made your introduction incoherent.
3- What’s the relevance of assessing stress oxidative markers? Add in your introduction how it’s implied in the pathogenesis of RA.
Methods
1- What was the age, sex and weight of the animals used in your model?
2- Line 112: you define group 1 as a vehicle control. However, it only received saline, while fustin was diluted in DMSO. Vehicle group should have received 5% DMSO. Group 1 is the control group (normal).
3- Line 122: you wrote arthritis modalités…you mean modalities?
4- Line 133: ‘Blood samples from the test animals were taken on day 22 using the retroorbital plexus and placed’, you should write ‘via (not using) the retroorbital’ if you talk about the anatomy… or ‘using the retroorbital plexus method’ if you talk about the blood collection method.
5- Line 138: You wrote ‘ALP,AST and AST’ you mean ALP, AST and ALT?
6- Line 141-146: revise tenses ‘precede, pour, add…’, all verbs must be in the past tense.
7- Line 151-152 is not good written. It should be paraphrased
8- Line 153: ‘Nrf2 and MPO’ should be in italic
9- Line 158-159: make sure the units used are correct ‘To 280 L assay buffer, 20 L tissue homogenate was added’??
10- Why didn't you use additional groups of animals receiving only fustin, to make sure that it has a regulatory effect, and does not affect balanced homeostasis?
Results:
1- Line 172 and 174: you talk about a significant effect. However, there is no significance shown on figure 1. Which statistical test did you use? According to your data and your graph, you should have used two-ways ANOVA repeated measures. Is it the case? Please add the significance signs (lettrers or *) to figure 1.
2- In all your figures, you compared the negative control to normal group, then the treated groups to the negative control. As the treated groups have shown significant differences in comparison with CFA administrated group, you concluded that fustin exhibited a promising anti-arthritis effect. However, it’s more relevant to compare your treated groups with normal group, because this is your reference since you didn’t use a positive control. It would be interesting to learn if fustin was able to revert CFA effect.
3- In general, reformulate your results using direct and concise writing. The current phrasing makes comprehension difficult.
Discussion:
In general, the findings are well discussed. Yet there are still some changes to be made:
1- line 192: ‘altered various haematological…’ instead of ‘alter’
2- Line 296-298: ‘Using a plethysmometer to measure paw thickness during CFA-induced arthritis is a well established and standardized method [34,35]. A plethysmometer was used to estimate the edematous hind paw of an inflamed rat in the current study’, should be moved to methods topic.
3- Line 320: you wrote: ‘Fustin treatment of arthritic rats significantly attenuate’, write ‘significantly attenuated’. Your work and the work reported in the literature are generally presented in the past tense. Do the same for the whole manuscript
4- Line 350: You wrote ‘Fustin restores the disparity between pro-oxidant and antioxidant markers, implying that it may play a role in retreating the destruction caused by free radicals.’ you can't talk about a restaurating effect since you didn't make sure there is no significant difference between your fustin treated goups and normal group.

Conclusion
1- Line 375: You wrote ‘Improvements in haematological and biochemical parameters may have a protective effect and thus accelerate disease progression.’ The meaning is paradoxical. Do you mean ‘and thus prevent disease progression’?

---

## Round 0.2 · accepted · Accept

Congratulation on your work.

Reviewer 1 ·

Basic reporting

The authors have responded to all the points mentioned previously in the review properly.

Experimental design

The authors have responded to all the points mentioned previously in the review properly.

Validity of the findings

The authors have responded to all the points mentioned previously in the review properly.

Reviewer 2 ·

Basic reporting

The reviewer's comments were properly addressed. I have no further comments.

Experimental design

No comment

Validity of the findings

No comment

Additional comments

No comment

Reviewer 3 ·

Basic reporting

The authors have improved the language quality of the writing, resulting in a well-structured, well-written, and unambiguous paper.

Experimental design

The experimental design and methods are well described for the reproducibility of the results.

Validity of the findings

The authors addressed all of our comments, thereby enhancing the validity of their findings. The data presented in the paper are statistically robust and well-controlled.